# A Low-Cost Approach to Crack Python CAPTCHAs Using AI-Based Chosen-Plaintext Attack

## Ning Yu *,† and Kyle Darling †

Department of Computing Sciences, College at Brockport, State University of New York,
Brockport, NY 14420, USA; kdar1@brockport.edu
* Correspondence: nyu@brockport.edu; Tel.: +1-585-395-5187
† Current address: 350 New Campus Drive, Brockport, NY 14420, USA.

**Abstract:** CAPTCHA authentication has been challenged by recent technology advances in AI. However, many of the AI advances challenging CAPTCHA are either restricted by a limited amount of labeled CAPTCHA data or are constructed in an expensive or complicated way. In contrast, this paper illustrates a low-cost approach that takes advantage of the nature of open source libraries for an AI-based chosen-plaintext attack. The chosen-plaintext attack described here relies on a deep learning model created and trained on a simple personal computer in a low-cost way. It shows an efficient cracking rate over two open-source Python CAPTCHA Libraries, Claptcha and Captcha. This chosen-plaintext attack method has raised a potential security alert in the era of AI, particularly to small-business owners who use the open-source CAPTCHA libraries. The main contributions of this project include: (1) it is the first low-cost method based on chosen-plaintext attack by using the nature of open-source Python CAPTCHA libraries; (2) it is a novel way to combine TensorFlow object detection and our proposed peak segmentation algorithm with convolutional neural network to improve the recognition accuracy.

**Keywords:** CAPTCHA security; authentication; open-source Python library; deep learning; convolutional neural network; TensorFlow

## 1. Introduction

CAPTCHA stands for Completely Automated Public Turing tests to tell Computers and Humans Apart. CAPTCHA was designed to restrict malicious attacks to their sites. CAPTCHA has also been referred to as a reverse Turing Test [1] since the CAPTCHA system is designed to determine if a remote request is from a human [2].

With the advancement of AI techniques, some AI problems may be solved with current AI technologies [3–5]. In the case CAPTCHAs are cracked by a hacker, the identification process will be abused and authentication can be bypassed by an automatic bot, which can further result in the catastrophic consequences to the website, service provider, or customers. Working on cracking CAPTCHAs can help identify those vulnerabilities of current CAPTCHA authentication and improve the system security.

In the past, methods of exploiting CAPTCHA to detect and extract text within an image ran through a disjoint process: localizing the whereabouts of the single characters within the image, segmenting and then recognizing the characters [6]. Then, attaching a dictionary or word processor to identify the possible text or dismiss any misleading or unlikely word. Modern approaches of creating such CAPTCHAs by text are to involve a high level of distortion that makes methods of localization, segmentation, and recognition rather difficult to perform. Such distortions are achieved

by overlapping and skewing certain portions of text which leads to segmentation becoming harder to apply towards cracking.

There are many schemes including proprietary CAPTCHA schemes such as Google reCAPTCHA and Yahoo CAPTCHA [7], and open-source CAPTCHA libraries such as Python Claptcha (https://pypi.org/project/claptcha/) and Captcha ( we differentiate **CAPTCHA** and **Captcha**. The uppercase one represents the authentication image while the latter one stands for the open-source library/module, https://pypi.org/project/captcha/). Claptcha-like CAPTCHAs are widely adopted for Internet service authentications by many profit or non-profit institutions such as Delta Airline and MITBBS. Since the proprietary CAPTCHA schemes were used by well-known companies such as Google and Yahoo, these types of CAPTCHA schemes have drawn more attentions in the security community and many existing CAPTCHA solvers aim to crack these proprietary CAPTCHA schemes [7,8]. It also results in that the proprietary CAPTCHA schemes add more techniques to increase their security. Complicated strategies such as exploiting the weakness of segmentation, lack of space, and increased overlap have been further added to CAPTCHA algorithms. Google's reCAPTCHA solution [9] offered a solution to analyze the cookies that might be left by the potential bots and determine if it is a potential threat. Based on a large scale image data testing, Microsoft proposed photograph CAPTCHAs named Asirra which was the first attempt of many potential tests approached by CAPTCHA Token Buckets to significantly decrease the yield of brute-force bots, while having a minimal effect on humans thus strengthening its security [10]. However, two years later this type of CAPTCHA was cracked by machine learning based attacks [11].

In contrast with proprietary CAPTCHA schemes, those open-source CAPTCHA libraries may have greater security risks to powerful learning machines. Our work have demonstrated a scenario where malicious learning machine can be trained infinitely through those tampered open-source libraries to perform such a chosen-plaintext attack. That is, a chosen-plaintext attack against CAPTCHAs can be performed by making the unlimited use of an encryption machine (open-source libraries) for generating well-labeled ciphertext (labeled CAPTCHAs) towards a malicious learning machine. This process is represented in Figure 1. The guaranteed performance of deep learning technology such as Convolutional Neural Network (CNN) [5,12] for character classification has injected new power into our work [4,5,13]. Our experimental results have shown that exploiting the open source nature can drive our Convolutional Neural Network to become a very well-trained model for hacking against Claptcha-like CAPTCHAs after the segmentation occurs.

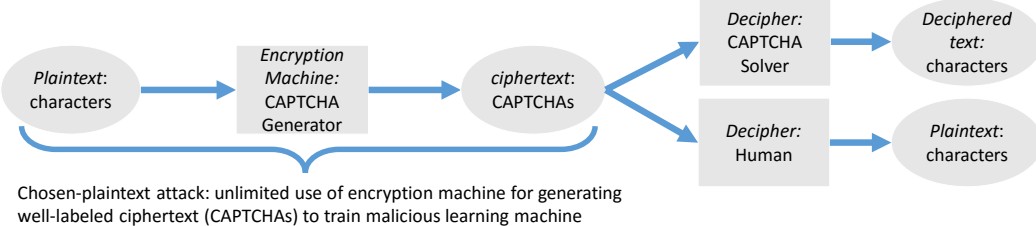

**Figure 1.** Chosen-plaintext Attack.

The rest of the paper is organized as follows. Section 2 is about the related work emphasizing on AI cracking techniques such as segmentation and recognition. In Section 3, our methods are described including the work flow, training, peak segmentation, TensorFlow object detection, CNN model, etc. Subsequently, in Section 4 the test results of our proposed framework are examined through local benchmark and external benchmark. At the end, we conclude the paper and discuss the future work in Section 5.

## 2. Related Work

CAPTCHA was defined as a cryptographic protocol whose underlying hardness assumption is based on an AI problem [14]. According to this definition, the CAPTCHA actually composes an AI-based encryption machine to encrypt some information so that either the CAPTCHA is not broken

and there is a way to differentiate humans from a computer system, or the CAPTCHA is broken by computer systems and a useful AI problem is solved [14]. Various Human Interactive Proofs (HIPs) [3] on the market including CAPTCHA were designed to tell computers and humans apart by posing challenges presumably too difficult for computers. Chellapilla and Simard found that pure recognition tasks can easily be broken by machine learning and a combination of segmentation and recognition tasks can make complex HIPs. They concluded that building segmentation tasks is the most effective way to confuse machine learning algorithms [3].

A variety of character segmentation techniques of general value were developed to attack a number of text CAPTCHAs [15]. Gabor filters offer the localization of spatial and frequency information in image signal processing and have been applied in CAPTCHA solving [16]. Log-Gabor filters were used to extract character components from CAPTCHA images along four directions while k-Nearest Neighbours were utilized for recognition [16]. Bursztein et al. tried to detect all possible cut points to segment a CAPTCHA into individual characters [17]. The potential cuts were found by examining the second derivative of curves. Based on the cut points, the meaningful potential segments may be extracted. The ensemble learning approach was adopted to vote the recognition scores, which is robust to noise [17]. Chen et al. classified the CAPTCHA segmentation methods into several categories such as projection, connection, width, contour, structure, and filter, and highlighted the character recognition methods based on neural network and deep learning tactics [7].

In addition to segmentation challenges, the current CAPTCHA solving methods are mostly based on known-plaintext attack where the number of training samples is limited. For example, Chen et al. collected 1000 CAPTCHA samples that contain 4000 characters total, limited by the availability of CAPTCHAs [5], where selective learning confusion class (SLCC) introduced a complex confusion class to improve character recognition. This type of known-plaintext attack has the limited capacity in training deep CNN models.

Due to the difficulty of having a large amount of labeled CAPTCHA data sets, Stark et al. proposed an active deep learning model that adopts only little training data set from the Cool PHP CAPTCHA framework for their CNN framework without any human intervention [18]. They described methods of segmentation and localization measures that allow for CAPTCHAs to be cracked and used for the training of a CNN [18]. These segmentation tactics applied in [19] were able to not only crack basic CAPTCHAs but also show the vulnerability in many other corporate level CAPTCHAs. Furthermore, CAPTCHA challenges developed by Google are often reused and may be more susceptible to being cracked [19].

Bostik and Klecka tried several machine learning methods to test a CAPTCHA gallery of 4950 synthetic characters generated by a PHP generator [20]. Their experimental results showed that a feed-forward network had better performance than k-NN, SVM, and other machine learning methods. The similar opinions were supported by many research that have validated the efficiency of neural network in cracking CAPTCHAs [4,5,13]. For example, two main stages, localization and recognition, were combined into their method in literature [4]. The former stage utilizes a heat map and k-means algorithm to determine if a character is located at the center with the support of an artificial neural network. The recognition stage creates a convolutional neural network in which results have proved it as an efficient method for recognizing the characters. The method also adopted BotDetect CAPTCHA, a paid and up-to-date service used by many government institutions and companies all around the world, for data training and testing.

Similarly, Jaderberg et al. proposed a method for text spotting from an entire image [6]. It computed a text saliency map by four state-of-the-art CNN classifiers in a sliding window where 16 different scales were repeated to target text heights. Text detection, character case-sensitive and insensitive classification, and bigram classification were performed through modified CNN networks [6]. CNN was invented to solve the document recognition problem [12] and the latest variants have been widely used for pertinent image classification [21]. Other models also could have been used to label the CAPTCHA images, such as Caffee, a deep learning framework that processes images and returns sets of labels being the most probable [21].

In order to relieve the dilemma of a limited amount of available CAPTCHAs for training CAPTCHA solvers, Ye et al. designed a complicated system that integrates Generative Adversarial Network as a synthesizer to simulate and generate a large amount of synthetic CAPTCHAs for feeding the CNN solver [13]. However, the input data are limited and this type of method is still based on the known-plaintext attack. In contrast, our proposed method based on chosen-plaintext attack can generate unlimited data sets theoretically for training.

On another side, in order to deepen the skill gap between human and computers for improving the existing CAPTCHA systems, Elson et al. presented novel theories [10] to give partial credit to a user's response. Usually computers score CAPTCHA responses with one bit of output: right or wrong. The intuition behind the Partial Credit Algorithm (PCA) is that a user-computer interaction is more important than right-or-wrong and user's response contains much more than one bit of information [10]. As many text-based schemes have been cracked, hollow CAPTCHAs and 3D CAPTCHAs emerged and have been deployed by large companies [8,22,23]. 3D CAPTCHAs rely on novel segmentation resistance by combining a crowding-character-together strategy and side surfaces of 3D visual effect. It leads to a promising usability even under strong overlapping between characters [23]. Sivakorn et al. provided a comprehensive examination of the design and implementation characteristics of the reCAPTCHA service and developed a low-cost attack that leveraged deep learning technologies for the semantic annotation of images [19].

Inspired by these related work above, we propose a novel approach for chosen-plaintext attack by utilizing those advanced techniques in previous research including CNN recognition network and segmentation techniques. In the following sections, our method and results will be described respectively.

## 3. Methodology

The methods were shown in Figure 2. The CAPTCHA generator was re-constructed based on the corresponding open-source libraries. We adopted two Python CAPTCHA libraries, Claptcha and Captcha. Characters in CAPTCHA were extracted according to the generated .csv data as shown in Table 1. Generating CAPTCHAs through the encryption machine without limitation, which is a feature of chosen-plaintext attack, can be used to feed the learning machine with numerous well-labeled training data. The learning machine was constructed on a customized Convolutional Neural Network (CNN) for recognizing characters. We used 42 symbols as the output class of CNN including 24 characters except O and I since the two characters O and I cannot be differentiated by a human from 0 and 1 effectively. It is a common strategy for a CAPTCHA generating system to exclude some particular characters such as O and I [13]. Our output classes also include 10 digits, and 8 special characters !@#$%^&∗. In addition, we used a novel way for the character segmentation by using TensorFlow object detection (TOD) in our CAPTCHA solver framework. Namely, we used TOD method to segment the characters and trained a CNN model to recognize characters. Furthermore, we combined TOD with our proposed peak segmentation method [24] to tighten the segmentation boundary and improve the recognition results.

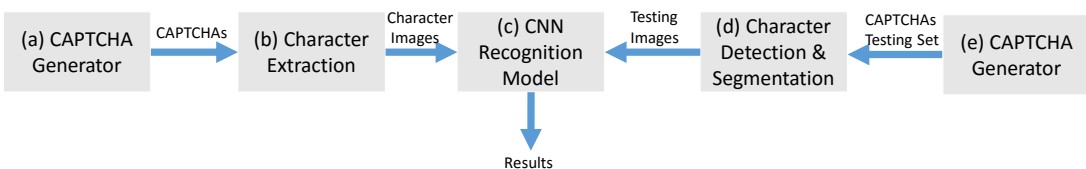

**Figure 2.** The work flow of our proposed framework. (**a**) Open-source libraries were modified to construct the CAPTCHA generator. (**b**) The generated CAPTCHA profiles were used to extract characters from the generated CAPTCHAs. (**c**) A CNN model was constructed for character recognition. (**d**) CAPTCHAs were generated for testing without saved profiles. (**e**) TOD and peak segmentation were used for segmenting the characters.

**Table 1.** CSV header for datasheet.

| Index | Image Location | Label | Height | Width | X | Y |
|:---:|:---:|:---:|:---:|:---:|:---:|:---:|
| Position | Path | List | List | List | List | Integer |

### 3.1. Generating CAPTCHA Training Samples

The first step in our process involved generating a large enough data set to train the neural network. In order to do so, the Claptcha module, an open-source Python CAPTCHA library, being used as a chosen-plaintext attack from Python must be manipulated in such a way to allow this type of generation to occur while keeping track of the character's profiles held within the image. We modified the Claptcha source code to contain the offset positions of the characters and return the text, the image, and the newly added offset positions to the program generating the samples. We carefully constructed a feed forward batch system with a data set restricted to only uppercase alphanumeric characters with exclusion to the letters I and O to not mistake as 1 or 0. The batch system is linked to our training model that can either accept the new input or continue off an existing data set, by having a batch size of 16,000 samples per generation.

Observing the headers as shown in Table 1, it becomes apparent that it is possible to capture all the relative positions of the characters within the CAPTCHA image. Figure 3a represents a data sample taken and displayed from this exact scheme of the data set, furthering the proof of concept about tracking the generation process for those CAPTCHA images.

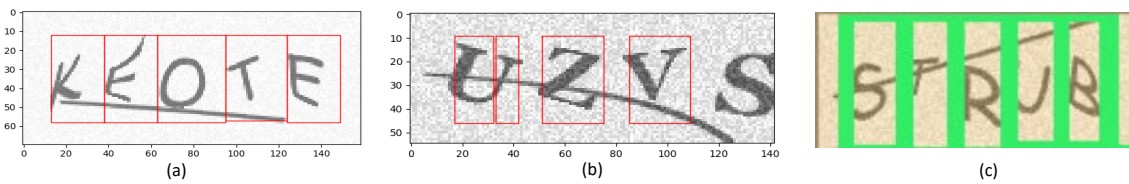

**Figure 3.** Examples of CAPTCHA Processing. (**a**) Modified Claptcha module provides characters location within image. (**b**) Bad segmentation attempt of the function. (**c**) Example of TOD Segmentation.

In recent literature [13], a complex Generative Adversarial Network (GAN) was constructed to produce fake CAPTCHA imagery for training purposes due to the limited number of real training data. It was also an example of using outside materials to train a Convolutional Neural Network to recognize CAPTCHA imagery. Compared with this complex GAN approach, generating CAPTCHA images from open-source modules is flexible, easily modifiable, and low-cost.

### 3.2. Peak Segmentation and Reconstruction Approach

Our characters generated by the Claptcha and Captcha modules provided an insight of the whereabouts of where that character may lie inside the image for the training process. However, as seen in Figure 3a the bounds are not tight enough, especially if each character had a wider space to reside within.

Inspired by the previous segmentation algorithms [4,7,24], we proposed our own peak segmentation approach that appears to be our best bet at generating large amounts of training data while keeping the bounding boxes relatively close to the character text. An example is shown in Figure 4 to visually observe how this peak segmentation process breaks down the image and reduces the bounds further. As it is displayed, the bounds of the original CAPTCHA image are tighter to the actual CAPTCHA letters compared with the bounds received from the Claptcha module. The pseudo code of *x*-axis peak segmentation algorithm is shown in Algorithm 1.

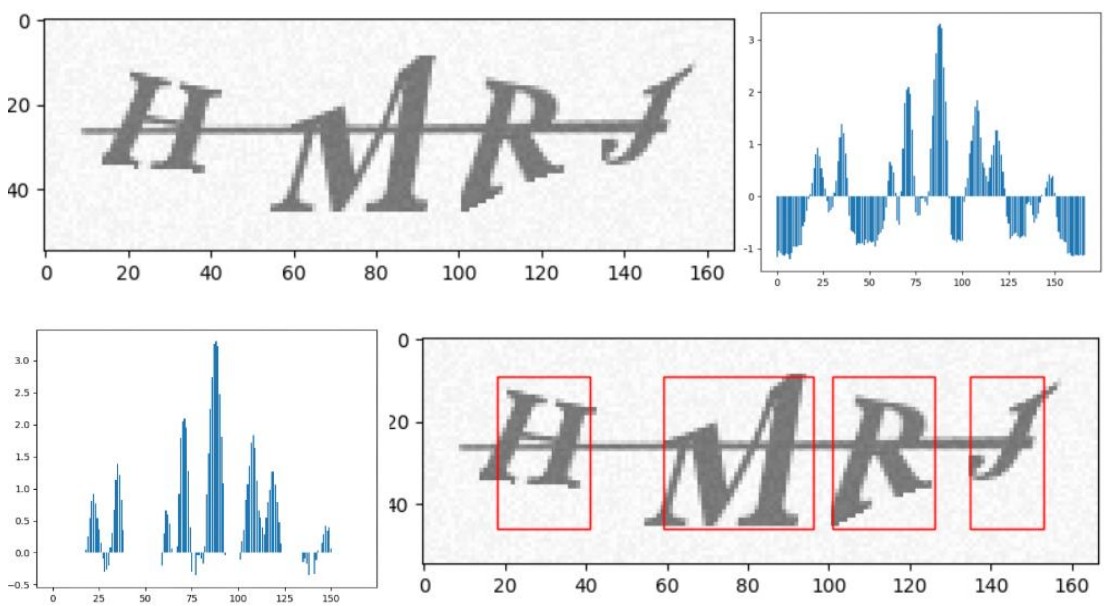

**Figure 4.** Locating and resizing bounds of CAPTCHA image.

Peak segmentation algorithm relies on the writing order of a CAPTCHA from left to right. Thus, the entire CAPTCHA is compressed into the *x*-axis by summing up the values in *y*-axis. The top right histogram in Figure 4 shows the truth where the horizontal line indicates the average value. After removing those below average by setting a threshold, the bottom left histogram in Figure 4 is able to show the approximate location of characters. Reflected in Algorithm 1, line 5 shows the initial process and lines 6–12 use a loop to scan *y* that saves values along *x*-axis. If the gap is larger than the user designated threshold as shown in lines 15–19, segmentation information is appended to a list named *character_indices*, the return value for *x*-axis segmentation. For *y*-axis segmentation, it is not shown here, which is much easier than *x*-axis segmentation since we assume only one character existing in *y*-axis after *x*-axis segmentation.

Even though the bounds have been tightened by peak segmentation, errors may occur through the segmentation where the same character can have two bounding boxes appear as shown in Figure 3b. When the bad split occurs, it directly causes the detection failed. We have also introduced another reconstruction function which takes in account for the number of segmentations that should occur, such that, if the number of segmentations is greater than the number of characters, then it reduces and combines smaller bounding boxes together. The reconstruction approach is used for testing the external benchmark.

### 3.3. TensorFlow Object Detection for Character Detection

TensorFlow object detection (TOD) approach is used as the segmentation component in our framework to detect character locations. Both TensorFlow object detection and peak segmentation are used in our framework. The former one is the primary approach for segmentation while the latter one is complementary to TensorFlow object detection.

Open-source TensorFlow detection model repository offers a magnitude of models, able to be trained further and integrated into image recognition work, in which faster regional Recurrent-CNN inception [25] was scored high in both accuracy and reasonable computation speed. Unfortunately, the preliminary test of the model and any other subsequent model displayed a drop out of being unable to identify the character within the image but was able to find the location a character resided. Thus, after integrating the pre-built model to detect characters, we segmented character images and extracted location information for feeding segmented characters to our recognition model. This approach ensures high accuracy by increasing a bit overhead towards benchmarking our two CAPTCHA modules.

---

**Algorithm 1:** Pseudo Code of *x*-axis Peak Segmentation for a CAPTCHA

---

  **Data:** CAPTCHA image converted into a binary image to be assessed for character locations. The image is then broken down into two parts, *x*-axis being positionally based and *y*-axis being a summed list of pixels dependent of the position. y is a list of sums of vertical pixels along *x*-axis direction; avg is the mean of y.

  **Result:** Output a list of positions that indicate the characters indices through a tupled list of segments $\triangle x$.

```
1  character_indices = []; // Controls the output list generated by segments △x.
2  start = 0; // Starting position on the x-axis to traverse.
3  reading = False;
4  count = 0;
5  // The below expression is to change all values in y to 0 if they do not meet
      the averaged value of calculated returns.
6  y = [i if i >= avg else 0 for i in y];
7  // The below loop iteration is to gather the character indices.
8  for index, i in enumerate(y) do
9      if i != 0 then
10         count = 0;
11         if not reading then
12             reading = True;
13             start = index;
14         end
15     else
16         count += 1;
17         // Threshold is a user designated value, defaulted at 3 pixels to stop
              reading.
18         if reading and count >= threshold then
19             reading = False;
20             character_indices.append((start, index));
21             count = 0;
22         end
23     end
24 end
25 // Character indices returned from loop iteration process.
26 return character_indices;
```

---

The TOD character detection model used on top of our character recognition model was built using Google object detection directory within the TensorFlow repository. The example of TOD Segmentation is shown in Figure 3c. Figure 5 illustrates the loss over a 10,000-iteration period for training this character detection model. We can view the loss rate upon training when we gave the same data received from our generator to our training model. The data give the detailed information such as where the character lies within the image and the label it has. In Figure 5, classification loss is the loss between the detection and a set of characters since it does not classify individual characters; localization loss is the loss of wrapping box around the character and its real location; RPN loss is the loss of combining the previous classification and localization losses; objectness loss is the loss of classification based on the bounding boxes; total loss is the total loss between classification, localization, RPN localization, and objectness losses; clone loss is the same as the total loss. In this project, we used the model generated from 10,000 iterations because the higher iterations can cause overfitting.

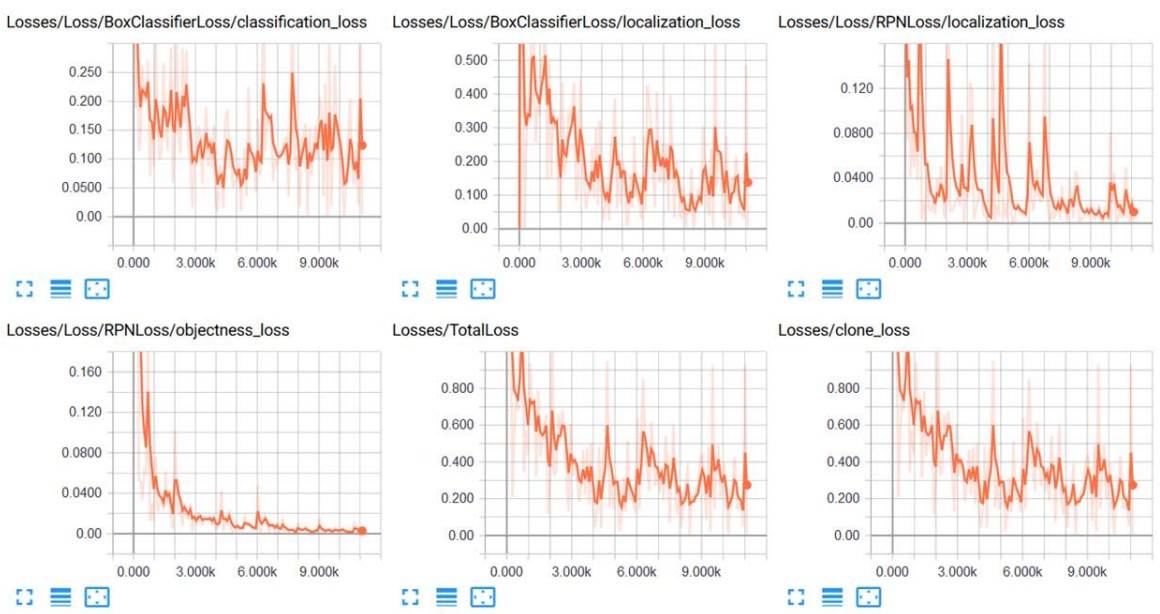

**Figure 5.** Loss over 10k iteration.

## 3.4. Convolutional Neural Network Model

We decided to create the CNN network in recognizing CAPTCHA characters according to the work of Kopp et al. [4]. We chose TensorFlow which allows us to construct a sequential model for our Network. Furthermore, it allows for direct access into creating a well-formed model prior to compiling the model.

First a constant input must be set in place for the network. A decided fix input size of a $28 \times 34$ pixel image was used, which allows the previous images generated to be segmented and reduced down to a $14 \times 17$ size. Being our input layer, the data passed in must be of dynamic nature and not all statically the same for our model to score effectively towards it. Using a Multi-Layer-Perceptron (MLP) rather than a Single-Layer-Perceptron (SLP) boosts the performance compared with the same scheme presented in [4]. Using this as a main basis of what to construct for our network, a Convolutional Neural Network model was built as shown in Figure 6. We adopted the batch size of 128, epoch of 40, and drop out rate of 0.2 [26]. The cross-entropy loss function is described in the generalized Equation (1),

$$L = -\sum_{c=1}^{K} \widehat{p}_{o,c} \log \widehat{p}_{o,c} \qquad (1)$$

where $\hat{p}$ is the probability of observation $o$ with class $c$ [27].

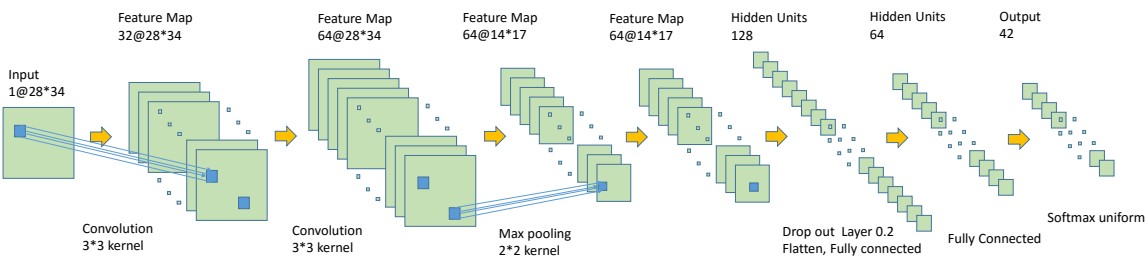

**Figure 6.** The architecture of a character recognition CNN.

The created CNN model features the following breakdown within the network itself of the sequential model. First a convolutional input layer was constructed using a ReLu activation layer of input shape 28 × 34 to measure the activation occurring within the standard inputs provided. Next, two convolutional layers applied with filters of 32 and 64 respectively were applied with using a ReLu activator, followed by applying a MaxPool2D layer in order to reduce the convolution down. We constructed another two layers of convolutional networks until we reached the Drop Out layer, given a 20% pass through rate. Then by flattening the layer, we reduced the layers down carefully by applying dense layering on top of it. Finally adding the last density layer by applying a SoftMax filter on top of it. The model described is of the best case found with previous trials conducted to receive various predicted and evaluated scoring accuracies generated by the sequential model. Previous attempts yielded accuracies around 80–90% but by adding an additional dropout layer and changing our last activation layer to a SoftMax activation, our model accuracy finished with a 99.32% training accuracy and 92.37% validation accuracy as shown in Figure 7. Compared to previous attempts, this model takes longer to get a higher accuracy but it consistently holds up to being able to correctly identify characters.

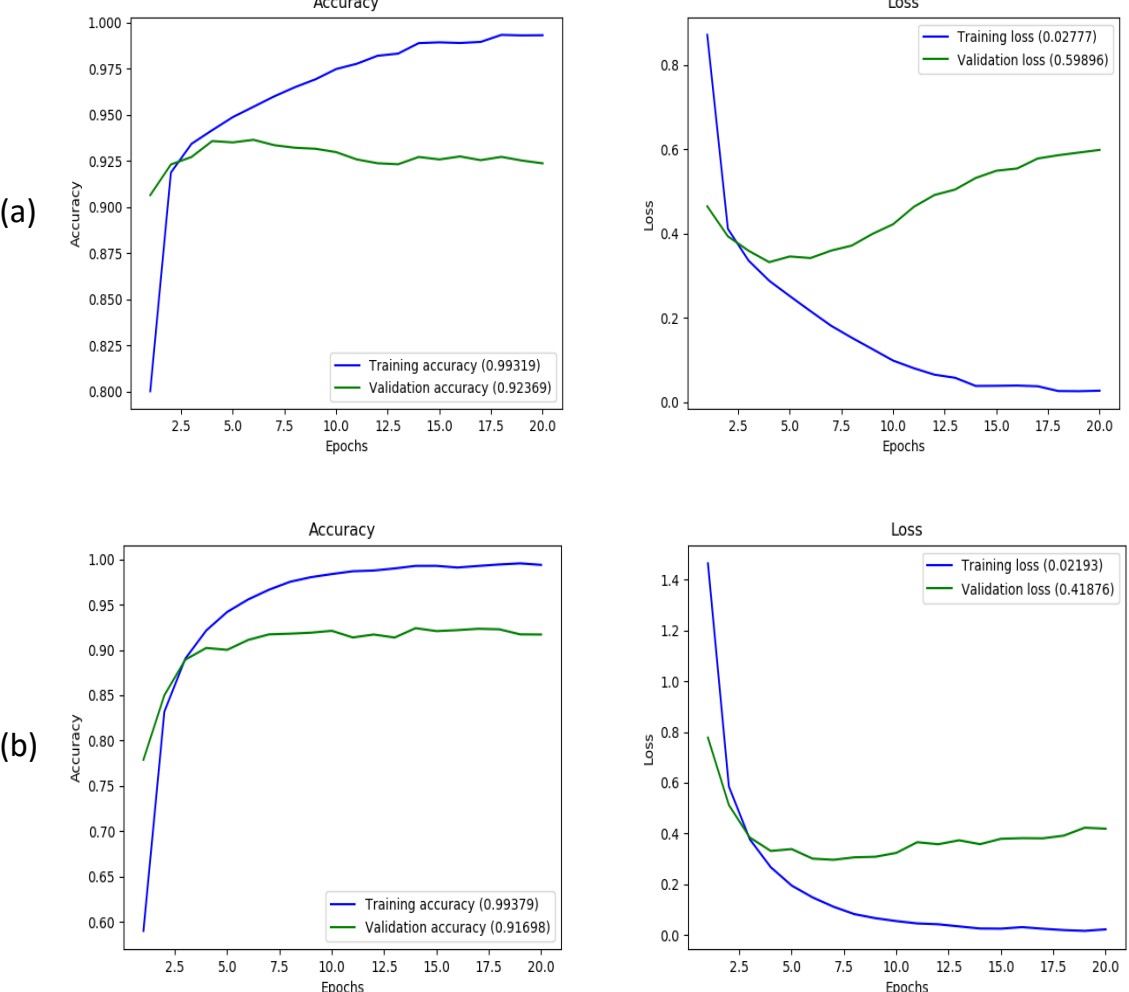

**Figure 7.** Accuracy and Loss of CNN model. (**a**) Training and validation results on CLAPTCHA data set. (**b**) Training model on both data sets. Both (**a**,**b**) were garnered from 20 epochs and trained on 60,000 samples.

## 4. Results

The test was performed over two benchmarks. One was generated locally from the two open-source CAPTCHA modules, named local benchmark; another was collected from the external resources to test its ability in cracking other CAPTCHAs, named external benchmark. The local benchmark was generated in a different way from the training procedure where CAPTCHAs' generating information such as positions, dimensions, etc. were saved. With respect to the local benchmark, all these information were not saved for testing.

Our model/framework was able to be trained on a basic personal computer (i7-6700k Quad-core 4 GHz CPU, Nvidia GPU GTX 1070 8 GB coupled with CUDA for TensorFlow processing, 16 GB RAM, 1 TB Seagate HD with less than 5 GB of free space). The time taken for a single batch from initial start was 15 min; hence the training of larger samples was all the more feasible for later testing. Though it is within reason, increasing the batch size causes more imagery to be stored in main memory and consumes more computing power per the larger batch.

The chosen CAPTCHA modules for evaluation came from two modules located within the pip module repository within Python3: (1) Claptcha—A module we used for modification and generation of data; (2) Captcha—A module that provides custom fonts to be applied in creating unique CAPTCHA strings. The two modules have their own unique differences that allow for CAPTCHA to be generated swiftly and evaluated within a reasonable time. For this, we modified the existing generating and training script into generating a Claptcha only model and a Captcha only model for benchmark comparisons along with combining the two modules together into a single model.

### 4.1. Local Benchmark

A test script was setup to focus on generating the two respective CAPTCHAs then running our two selected modules on top of them to evaluate overall performance. We did this by first generating 200 samples of each CAPTCHA module and then ran our framework on the two sets of generated samples. All CAPTCHAs in the local benchmark were simply generated through the test script without saving any information such as positions, dimensions, etc. for testing.

Figure 8 shows evaluation results for our conjoined method which display vital information for key aspects needed in cracking CAPTCHAs. We notated the proposed method as TOD+CNN that means TOD for segmentation and CNN for character recognition. Also, we compared TOD+CNN with Peak+CNN (peak segmentation and CNN) since this project is the first work on cracking the open-source Python CAPTCHA data set and there is no other existing results that could be used for comparison. Figure 9 gives the results of the true/false positives over the character-level accuracy. It becomes apparent in Figures 8 and 9 that we can take away a few key aspects: (1) Increasing character length tends to cause degradation in cracking CAPTCHAs. (2) TensorFlow Object Detection outperformed the Peak Segmentation framework. (3) There is no clear character(s) that are consistently misclassified by our model.

It is worthwhile to look into that if we have our conjoined modules being evaluated, we should evaluate the individual modules as well. This allows us to detect if any common interest or help is happening or degradation occurring by combining the two modules together. Figure 10 shows two examples taken when we ran the individual training on the same benchmark as above but only on a 6-character CAPTCHA scheme.

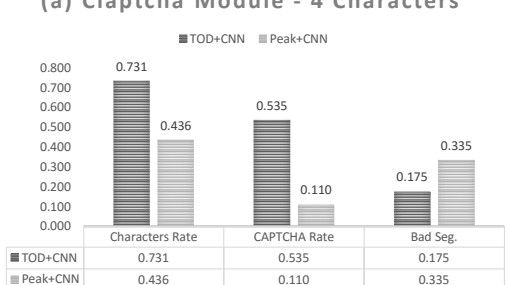

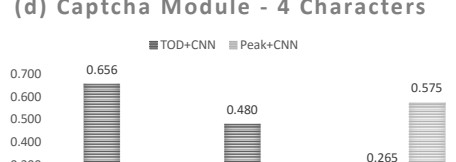

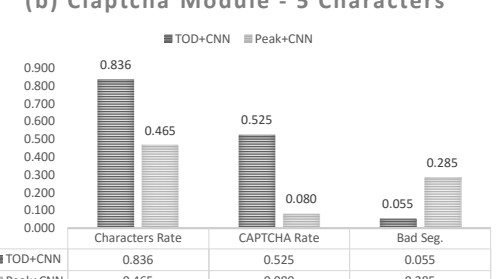

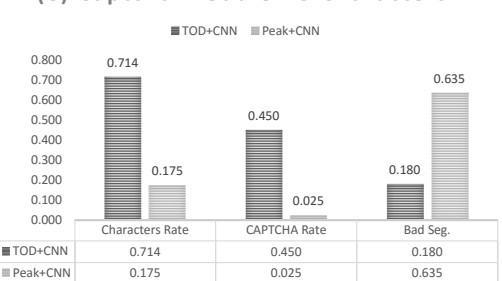

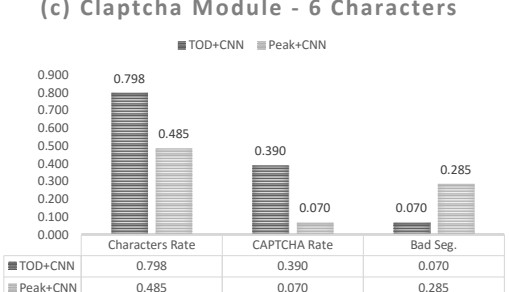

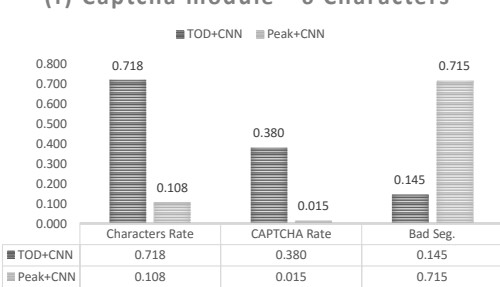

**Figure 8.** Test Results for Cracking Rate over Different Types of CAPTCHAs.

Figure 10 indicates an interesting relationship and it rules that: (1) Combining the two modules (Claptcha and Captcha) together helps increase character and CAPTCHA crack rates. It is clear that the individually trained modules have the inferior results compared with well-trained modules. (2) TOD+CNN still outperforms Peak+CNN in most cases.

We can further conclude that generating the two modules and combining them into a single model does not infringe on performance but rather works together to solve each other's respective characters. With this knowledge in hand, we can extend the test on a separate, external, and unknown CAPTCHA imagery.

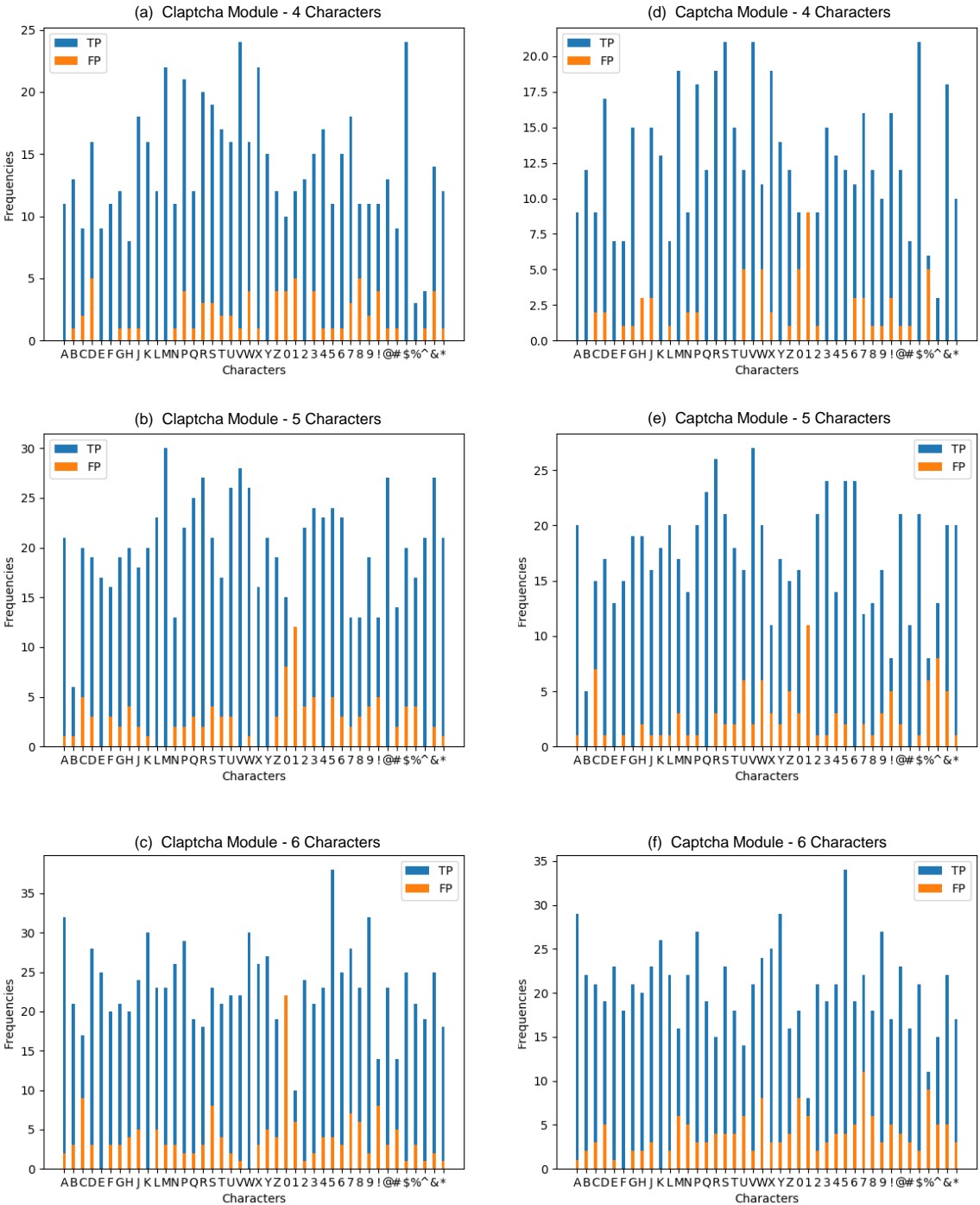

**Figure 9.** Statistics of True Positive and False Positive over Different Types of Testing CAPTCHAs.

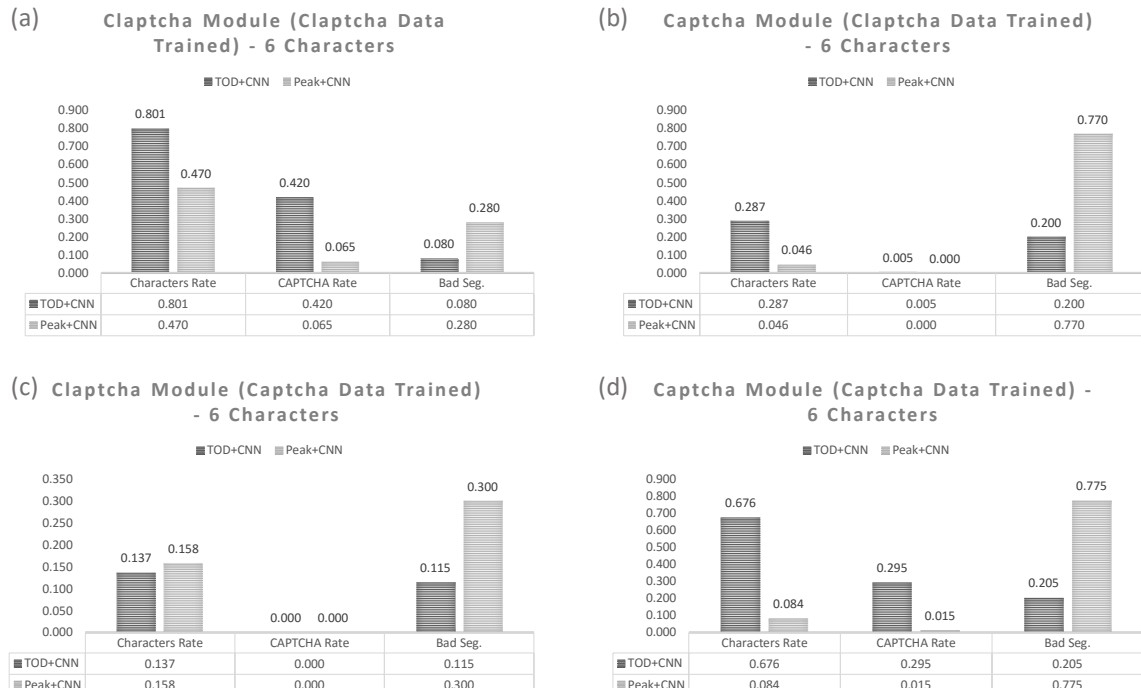

**Figure 10.** Test Results of Individually Trained Modules. (**a**,**b**) Only Claptcha data are trained to validate Claptcha and Captcha modules respectively. (**c**,**d**) Only Captcha data are trained to validate Claptcha and Captcha modules respectively.

## 4.2. External Benchmark

To gain a better understanding as for the extensive ability of our proposed model, we focus on testing our model on outside resources. We collect two typical data sets. One is the data set we extracted from HashKiller, a well-known hacker website for security research (https://hashkiller.co.uk/Cracker/MD5, as far of Match 2019. The HashKiller website recently switched to reCAPTCHA since they realized the vulnerabilities attached with using old style plain-text CAPTCHAs. The original website can be found using the wayback machine https://web.archive.org/web/20180401000000*/https://hashkiller.co.uk/ which contained CAPTCHAs our framework cracked.). We were able to manually gather 13 CAPTCHA images of length 6. The type and style of this data set are different from the Claptcha and Captcha as shown in Figure 11a, which was used for testing the extensive ability of our model. We name it HashKiller13 thereafter. Another external benchmark is Claptcha-like data set, which was extracted from an Internet service authentication used by Delta Airline. It contains 40 CAPTCHAs with length of 5, composed of digits only as shown in Figure 11e. We name it Delta40 thereafter. The generating algorithms of HashKiller13 and Delta40 are unknown to our test.

Running our model combined with TOD+CNN framework, we were able to crack 1 CAPTCHA of length 6 out of 13 in HashKiller13 data set as shown in Figure 12. When observing the data, we found that our framework failed at segmenting certain character sets. To increase performance, we could hypothesize that in the case of Figure 11b that a peak segmentation would split the two apart evenly. This means that if we ran our TOD framework then ran peak segmentation afterwards to further reduce the imagery, we might have some success. Upon testing this hypothesis, it was found to be correct in the sense of improving accuracies as shown in Figure 11c,d where peak segmentation does help improve accuracy but often leads to overcorrecting the image and often segmenting portions not needed. Tweaking the threshold of Peak Segmentation from 3 pixels to any range from 5 to 10 led to barely any improvement. Thus it is concluded in our test that our TensorFlow object detection is in

fact the strong framework but an optimized peak segmentation combined with TOD would in fact improve segmentation performance over the HashKiller13 benchmark.

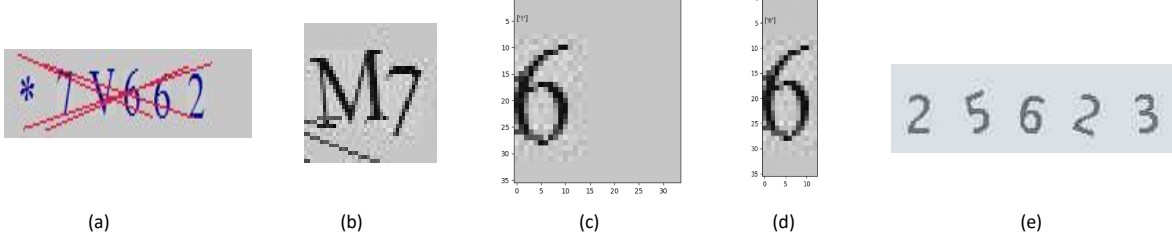

(a)      (b)      (c)      (d)      (e)

**Figure 11.** Examples of External Benchmarks. (**a**) An example of external benchmark HaskKiller13. (**b**) TOD failed to segment the two characters. (**c**) Result after TOD segmentation. (**d**) Result after the combined method. (**e**) An example of external benchmark Delta40.

Delta40 shows Claptcha-like patterns that can be easily cracked by our model. As shown in Figure 13, Delta40 was solved in high accuracy by our model combined with TOD+CNN framework. TOD+CNN has the better performance than TOD+Peak+CNN. Therefore, it means that TOD+Peak is not always a good idea for cracking CAPTCHAs while TOD+CNN is really a strong method for object segmentation over Delta40 benchmark. Such high cracking accuracies with more than 90% for character-level accuracy and more than 75% for CAPTCHA-level accuracy expose great security vulnerabilities in such CAPTCHA-based Internet service authentication for Delta Airline.

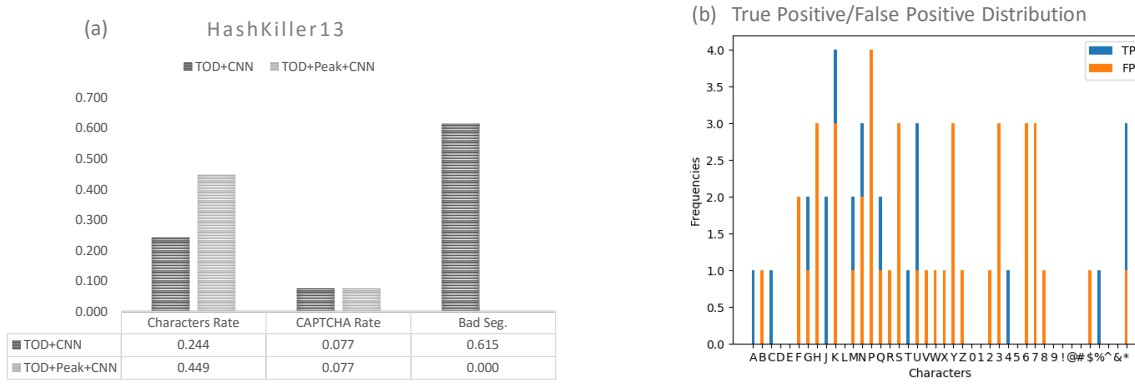

**Figure 12.** Test Results over External Benchmark HashKiller13.

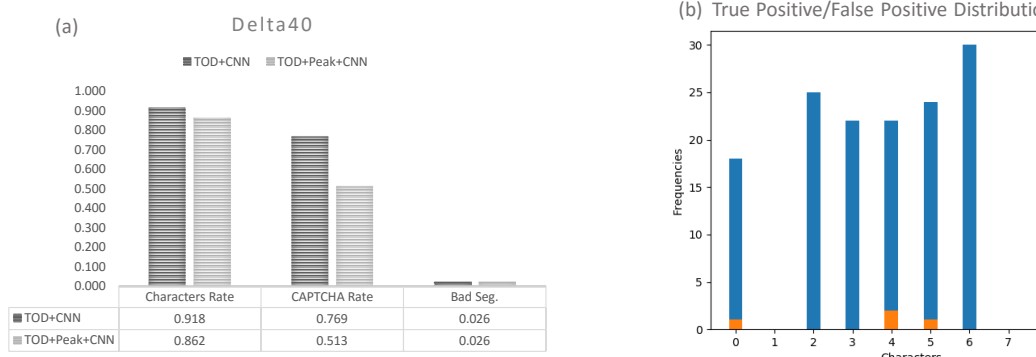

**Figure 13.** Test Results over External Benchmark Delta40.

## 5. Conclusions and Future Work

Python contains a large amount of vastly diverse modules to aid the development and security of a corporation's backend infrastructure. By abusing the open-source nature of the Claptcha and Captcha modules, we were able to generate thousands of data sample images to be trained upon a well-formed CNN in an effort to break any Claptcha and Captcha based security measures. Following the intuition, we applied our framework to crack other additional CAPTCHAs in order to test the extensive ability of our model. Hence we have created a string of methods that eventually lead to breaking a CAPTCHA-based security implementation for that specific case.

Our proposed method also is low-cost because (1) utilizing an open-source library itself is inexpensive for the hacking software development; (2) unlimited dataset can be generated through the revised software, which greatly decreases the training cost and efficiently increases the accuracy of modeling; and (3) the entire method and framework can be implemented in an economically affordable personal laptop.

In this paper, we generated a large number of samples from the open-source Python CAPTCHA modules and modified them in such a way to track the character's profiles placed within the image. We segmented the generated images and trained a customized CNN for those segmented sample characters. Also, the character's boundary can be reduced by peak segmentation. Provided by a process of TensorFlow object detection, we were able to carefully detect which character is present in a segmented sample. The validation and evaluation results have shown that the well-designed TOD+CNN model has the ability to crack open-source CAPTCHA libraries such as Python Claptcha and Captcha and it has the extensive ability to crack external Claptcha-like CAPTCHAs such as Delta40 benchmark. Also, it has shown that TOD+CNN has a certain ability to crack other type of CAPTCHAs such as HashKiller13.

We conclude that the main contributions of this paper rely on two thrusts. (1) We proposed the first low-cost chosen-plaintext attack method by taking advantage of the nature of open-source CAPTCHA libraries. (2) It is a novel way to combine TensorFlow object detection with a convolutional neural network to improve the accuracy of cracking CAPTCHAs. Certainly, our method also has some drawbacks that need to be improved in the future. For example, in Delta40 benchmark our model failed to detect digit 8 since it looks like character B; in the local benchmark, it failed to recognize digit 1 since it looks like character L. Meanwhile, we plan to optimize the segmentation algorithms, particularly to explore new methods to improve the problem of bad split that can greatly reduce the performance of designed models. We also plan to further enhance the CAPTCHA-cracking procedure and form a generic framework by exploiting the nature of any open-source library.

Deep learning technologies have emerged as a new propeller to extract semantic information from images and drive more novel low-cost attacks against CAPTCHAs. Creating new CAPTCHA schemes and their alternatives as well as exploring their flaws are developing into a new research direction [28]. The future of CAPTCHA will greatly rely on the exploration of novel directions [19,29–34]. When CAPTCHA was invented, Von Ahn et al. pointed out that not all hard AI problems can be used to construct a CAPTCHA, similar to computational problems: not all hard computational problems yield cryptographic primitives [14]. In order for an AI problem to be useful for security purposes, there needs to be an automated way to generate problem instances along with their solution. CAPTCHAs should not base their security in the secrecy of a database or a piece of code without advancing the algorithmic state of the art [14]. Actually, these statements imply that the next generation CAPTCHA system should be equipped with more advanced AIs to face with the challenges from AI cracking.

**Author Contributions:** conceptualization, N.Y.; methodology design, N.Y. and K.D.; implementation, K.D.; validation, K.D. and N.Y.; writing and editing, N.Y. and K.D. Please turn to the GitHub for the project source code and data sets.

**Funding:** This research was funded by School of Art and Sciences and the Research Foundation for SUNY, The College at Brockport.

**Acknowledgments:** We are grateful of the support from the Department of Computing Sciences and School of Art and Sciences, The College at Brockport, State University of New York. Authors also are grateful of suggestions, comments, and editing from Christine Wania.

**Conflicts of Interest:** The authors declare no conflict of interest.

## Abbreviations

The following abbreviations are used in this manuscript:

| | |
|---|---|
| AI | Artificial Intelligence |
| CAPTCHA | Completely Atomated Public Turing test to tell Computers and Humans Apart |
| CNN | Convolutional Neural Network |
| DL | Deep Learning |
| HIP | Human Interactive Proof |
| kNN | k Nearest Neighbor |
| ML | Machine Learning |
| TOD | TensorFlow Object Detection |

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
