# Peer review of "A Low-Cost Approach to Crack Python CAPTCHAs Using AI-Based Chosen-Plaintext Attack"

_applsci, doi:10.3390/app9102010_

Round 1

Reviewer 1 Report

The authors propose a deep learning approach to solve CAPTCHAs for the specific open sources libraries Claptcha and Captcha. Specifically, they exploit the openness of these libraries to create an “unlimited” size of labeled CAPTCHA samples in order to train the detection algorithm. Furthermore, they modify the libraries in such a way that the libraries will also output the offset position of the character (location) within the CAPTCHA image. They follow a chosen-plaintext attack approach, as they choose the text input of the libraries and they compile a dataset of labeled input-output pairs.

For the purpose of detection, they utilize a combination of TensorFlow object detection and peak segmentation algorithm with convolutional neural networks (CNN). The first two methods are utilized for the segmentation of the characters. Tensor object detection is provided by the well-known TensorFlow library, while peak segmentation algorithm is a contribution of their own.

The presented idea is interesting and definitely, it will affect negatively businesses and organizations relying on the specific libraries for hindering the access to their websites by automatic bots. However, the manuscript requires certain major revisions in order the authors’ arguments will be clear and comprehensible to the potential readers.

As the focus is on the specific open source libraries and not on proprietary CAPTCHA schemes, they should justify the popularity of this subject. Namely, they should state how common is the usage of these open libraries (i.e., Claptcha and Captcha).

In the introduction section, the authors should state what would be the (negative) effects to the website owners in the case their CAPTCHA mechanism is solved. What will be the security consequences? A more detail explanation of the motivation (why is it worth to deal with this concept) is required.

At the end of the Introduction section, they should provide the structure (organization) of the publication.

The related work section should be organized according to the concept of each work and presented in chronological order.

Regarding the methodology, it is not clear how the exclusion of the low case characters and the alphabetic characters O and I will affect the accuracy of the solver. How these specific characters will be recognized at the end? Since they state, they utilize as output class of the CNN, 24 alphabetic characters, 10 digits, and 8 special characters. So, I am wondering how the detector will behave in the case the CAPTCHA contains other characters than the one considered.

The peak segmentation algorithm, which is a contribution of their own, is not explained adequately in section 3.2. There, they should describe the presented (pseudo) algorithm in more details.

As concerns the evaluation of the proposal (section 4), it is not clear, if they evaluate the accuracy of the solver with the modified (position output) libraries or with the standard source code. Also, it will be useful to apply the detection mechanism to other datasets collected from the web and possibly created by a different library. This way, we can deduce the generality of the solver.

They have to justify the “low-cost” characterization of the proposal. To this direction, it will be helpful to measure the computational performance.

The usage of the English language needs improvement in order to increase the comprehension of the paper.

Author Response

Dear Reviewer,

Thank you so much for reviewing my paper. We made a major revision and wrote the response letter according to these valuable comments.

Please kindly check out the uploaded letter.

Sincerely,

Ning Yu

Reviewer 2 Report

Line 20: could read:  "...from a human...."

Line 33: never heard of the word "askewing", perhaps use "skewing" instead?

Line 38: "....other techniques...."     and "....offer a solution....."

**** There are various other issues with the English in this paper, but these are too frequent to list them all. Hence, I will concentrate on the content, and recommend that the paper is properly proofread by a native English speaker before potential publication ****

Line 45: Citation required for CNN.

Lines 50/51: Citations for Yahoo / Python Captchas.

Line 55: "unlimitedly ?".

Line 64: Ref's required for "various HIPs..."

Line 98: "large amount OF....."

Line 110: "a bunch of...." seems inappropriate - use of slang. 

Line 242/243: More detail on the setup would be useful here.

Overall, the approach is sound, and the paper is well structured, with the results generally well presented. Some elements of the presentation (grammar, etc) might be improved to further polish the paper. In summary, a decent, stimulating piece of work, which should be of interest to the intended audience.  

Author Response

Dear Reviewer,

Thank you so much for reviewing my paper. I made a major revision and wrote a response letter according to reviewer's comments. 

Pleas kindly check out the uploaded response letter. Thank you very much.

Sincerely,

Ning Yu

Round 2

Reviewer 1 Report

I will like to thank the authors for their effort to revise their manuscript according to the reviewers’ comments.

The paper can be accepted for publication given the following minor corrections:

Concerning section 4.2 (External Benchmark), the provided link for HashKiller dataset directs to hash lookup service (cracked hashes). Nowhere on the website refers to CAPTCHA. Therefore, please correct the link and revise the statements accordingly. Furthermore, explain with which algorithm are the external CAPTCHA datasets generated (i.e, HashKiller13 and Delta40). Is it used any proprietary algorithm or open source library?

Regarding the numbering of figures, tables, etc, please place a space between the word and the number, namely

Figure1 --> Figure 1

Table1 --> Table 1

Author Response

Dear Reviewer,

Thank you so much for reviewing our manuscript. We made a minor revision following the reviewer's comments.

Sincerely,

Ning Yu
